# Incidence of Leader–Member Exchange Quality, Communication Satisfaction, and Employee Work Engagement on Self-Evaluated Work Performance

**DOI:** 10.3390/ijerph19148761

**Published:** 2022-07-19

**Authors:** Zuleima Santalla-Banderali, Jesús M. Alvarado

**Affiliations:** 1School of Psychology, Universidad Espíritu Santo, Samborondón 092301, Ecuador; 2Psychobiology & Behavioral Sciences Methods Department, Faculty of Psychology, Complutense University of Madrid, 28223 Madrid, Spain; jmalvara@ucm.es

**Keywords:** leader–member exchange quality, communication satisfaction, employee work engagement, work performance, positive organizational psychology

## Abstract

Within the scope of the Theory of Demands and Labor Resources, the Healthy & Resilient Organizations (HERO) Model, and the Leader–Member Exchange (LMX) Theory, this research contrasts a mediation model in which evidence on the factors that affect work performance is integrated, thus establishing the direct and indirect relationships between LMX quality, communication satisfaction, employee work engagement, and self-rated work performance. A total of 488 workers participated in this research. Adequate goodness of fit was found in the model (χ2 = 3876.996, *df* = 3715, *p* = 0.031; χ2/*df* = 1.044; CFI = 0.999; TLI = 0.999; SRMR = 0.056; RMSEA = 0.010): the LMX–work performance relationship is mediated by communication satisfaction and work engagement, whereas the LMX–work engagement relationship is mediated by communication satisfaction. This has led to the conclusion that, as employees consider the relationship with their superiors to be of higher quality, the satisfaction they experience in terms of organizational communication increases, and as organizational communication satisfaction increases, the extent to which employees feel more vigorous, involved and concentrated, and absorbed by work also increases, which, in turn, leads them to perceive their work performance to be higher.

## 1. Introduction

Although there is no univocal answer to the question, “What is an organization?,” following the basic notions of the Systems Theory, the Institutional Theory, and the Human Relations Theory [1,2], organizations can be conceived as social systems created and shaped by people who, in some way, intend to work towards achieving similar or common objectives [1,2,3]. These systems are structured in subsystems that must be coordinated in order to achieve organizational goals [1,2,3]. Therefore, following the foundations of the Human Relations Theory [2], the performance of any organization depends fundamentally on the behavior and the interrelationships that occur between its members, and between them and stakeholders. In this context, one of the challenges of organizations is to set up conditions to maximize the performance of their members [4]. For this, it is essential to understand what are the factors that determine the optimal functioning of people at work [5]. In the search for those factors, one of the approaches that researchers have incrementally adopted since approximately 1995 [6], and that has generated broad and relevant knowledge from a theoretical and practical perspective, is Positive Organizational Psychology. The objective of this approach is to describe, explain, and predict the optimal functioning of individuals and groups within organizations [7], emphasizing their positive aspects, strengths, and capabilities, rather than their weaknesses or negative aspects [6].

Under this approach, and based on the Job Demands–Resources Theory [8], the Healthy & Resilient Organizations (HERO) Model [9], the Leader–Member Exchange Theory, and empirical evidence on the factors that affect work performance, this research proposes and empirically contrasts a mediation model according to which the relationship between leader–member exchange quality and self-rated work performance is mediated by communication satisfaction and employee work engagement, whereas the relationship between leader–member exchange quality and employee work engagement is mediated by communication satisfaction (Figure 1).

In this way, the results of this study contribute to broadening the available knowledge regarding the aspects that affect employee work performance, and how such factors relate to each other, thus integrating, in a single model, variables whose relationships have only been studied independently. Therefore, this study provides insight into the complexity underlying the prediction of work performance. From a practical point of view, the results of this study serve as a basis for decision-making, hence guiding actions aimed at developing healthier organizations that maximize the performance of their employees.

### 1.1. Work Performance

Since the 1990s, work performance has been understood as something that goes beyond work itself [10]. It has been defined as the behaviors, rather than the outcomes (i.e., the consequences of behaviors) of the members of an organization that are relevant to the achievement of its objectives [11,12,13,14,15,16,17], and that are under the control of individuals [16,17,18]. Likewise, there is consensus that the “work performance” construct is multidimensional [15,16], with a hierarchical structure consisting of a general factor at the top of the structure, followed by several dimensions at the lower levels, which measure different performance aspects [14,19,20,21].

The Job Demands–Resources Theory [8] is aimed at describing, explaining, and predicting work performance and employee welfare, and it has served as the basis for several empirical studies. According to this theory, understanding work performance requires considering: (a) job resources (i.e., the physical, psychological, organizational, or social aspects of work, such as social support, supervisor support, autonomy, feedback, development opportunities, task variety, etc.) and personal resources (i.e., positive self-evaluations regarding the perception of one’s ability to control and influence the environment, such as self-efficacy, self-esteem, optimism, etc.); and (b) job demands (e.g., work pressure, overload, emotional demands, unfavorable physical environment, etc.). Job demands predict variables such as burnout or employees’ psychosomatic health problems, whereas the resources predict variables such as job satisfaction, motivation, organizational commitment, and engagement. Such variables, in turn, affect work performance [8]. Thus, according to this theory, aspects such as work engagement mediate the possible relationships between job and personal resources and work performance, as well as between job demands and work performance.

In line with the Job Demands–Resources Theory [8], but focused on healthy organizations, the HERO model [7,9] states that obtaining healthy organizational outcomes, such as high employee performance, depends on: (a) the extent to which an organization uses healthy organizational resources and practices, understood as task and interpersonal resources (e.g., task clarity, autonomy, supportive climate, teamwork, feedback, positive leaders, etc.), and strategies that allow structuring and organizing work (e.g., adequate organizational communication); and (b) the extent to which the members of an organization are healthy in terms of self-efficacy, mental and emotional competences, organizational-based self-esteem, confidence, positive emotions, job satisfaction, and work engagement [7,9]. These two components relate to organizational outcomes both directly and indirectly, in the sense that healthy organizational resources and practices predict the aspects that characterize healthy employees, and, in turn, those aspects predict healthy organizational outcomes [9].

### 1.2. Leader–Member Exchange Quality

Both the Job Demands–Resources Theory [8] and the HERO Model [7,9] highlight the crucial role that leaders or supervisors have in the behavior of their supervisees. This is because: (a) leaders are the ones who have the formal authority to indicate what must be done and how it should be done, and, at the same time, their own performance depends on that of their team members; (b) leaders act as role models [22]; (c) one of their tasks is to manage their followers’ affection, which affects performance [23]; and (d) their leadership style can influence constructs such as employee work engagement [7,9,24].

One of the relevant aspects in terms of the relationships established between leaders and each of their supervisees is exchange quality (work and social interactions that are established between both parties) [25,26,27]. In this sense, under the framework of the social exchange theory, the leader–member exchange theory states that, since both leaders and their supervisees have a limited amount of personal, social, and organizational resources, leaders selectively allocate such resources among their supervisees [27,28,29], thus establishing different relationships and exchanges with each of them [28,29,30].

The exchanges that characterize low-quality leader–follower relationships (“out-group” relationships) are based on the formal requirements contractually established [28,31,32,33,34]. In this type of exchange there is more formal supervision under more closed communication systems, in which leaders use their authority and provide less support, trust, and attention to their supervisees [29]. For this reason, in low-quality LMX, employees have fewer opportunities to influence decision-making [29].

In contrast, high-quality LMX (“in-group” relationships) is characterized by going beyond the formal contract [31,33,34]. Such relationships present high-quality communication [35] and bidirectional information exchange in more open communication systems. Likewise, there is a high-level of organizational and personal resources exchange, and there is mutual trust, sympathy, consideration, professional respect, cooperation, and loyalty between parties [29,30,33,34,36,37,38,39]. In high-quality LMX, leaders provide their followers with more social support and autonomy, assign them more desired tasks, and give them more opportunities for training and development [31,32,40].

Since in high-quality LMX leaders treat their subordinates more favorably [25,37], such employees have advantages that those who maintain low-quality LMX do not have. Consequently, given the principle of reciprocity of the Social Exchange Theory, it is expected that, under these conditions, employees respond by expressing more positive attitudes towards work, feel more optimistic and self-efficacious, show greater engagement [41], and have higher levels of task and contextual performance [31,34,37,40,42]. In this sense, the results of different research indeed show that LMX quality is positively and significantly correlated to the performance evaluations carried out by supervisors [37,39,42]. Particularly, LMX quality is positively correlated to task performance, organizational citizenship behaviors [28,34,39], and contextual performance [37]. On the other hand, a negative relationship has been found between LMX quality and counterproductive work behavior, both self-evaluated and evaluated through objective indicators [34]. However, Greguras and Ford found that task performance evaluated by supervisors was only predicted by the loyalty dimension, and that none of the construct dimensions predicted organizational citizenship behaviors [33].

Moreover, different studies have shown that the relationship between LMX and work performance is not only direct, but that it is mediated by employee work engagement: LMX quality directly and significantly predicts employee work engagement [30,31,32,38,43], and work engagement significantly predicts in-role performance [32,43], contextual performance [43], and overall work performance [31,38]. Kim and Koo found that LMX had a significant indirect effect on work performance through the direct effect it has on job engagement, which predicts organizational engagement, which is directly related to work performance [30].

### 1.3. Organizational Communication

Besides leader behavior, the HERO Model [7,9] includes organizational communication as one of the practices on which the performance of the members of an organization depends, and, according to the Leader–Member Exchange Theory, communication is also one of the key aspects associated with leader–member exchange quality. Thus, in both theoretical perspectives it is recognized that communication is the essence of any social system [44], and that it is one of the most important processes in any organization [45,46], and one of the main factors that determine its subsistence and success [47,48,49,50,51], since it is what allows an appropriate coordination between the different units that make it up to ensure the actions carried out are consistently aimed at achieving its objectives [44,48,52,53].

Likewise, “good” supervisors must adapt their communication to efficiently and effectively transmit the vision, mission, and objectives of the organization [54]. Indeed, the ability of supervisors to get employees actively involved in achieving organizational goals, making them seek for better performance, depends on how the communication process takes place [37,48,50]. In fact, a positive and significant correlation has been found between perception of communication flow with management and in-role and extra-role performance [50], and between supportive supervisor communication evaluated by collaborators and task and contextual performance evaluated by supervisors [37]. There is also evidence that internal organizational communication and internal supervisor communication significantly and positively predict employee engagement [55].

Effective organizational communication leads the members of an organization to feel satisfied with such communication [37]. Communication satisfaction can be conceptualized as an individual’s satisfaction with the whole communication environment at the interpersonal, group, and organizational level [29,46,51].

Regarding the elements that relate to communication satisfaction, it has been found that the higher the LMX quality, the greater the communication satisfaction in interpersonal, group, and organizational contexts [29]. On the other hand, proper communication allows supervisors to maintain an effective relationship with their subordinates, thus contributing to the achievement of desirable organizational outcomes [54], such as better productivity, performance, and external customer orientation [56]. In fact, some research has shown a direct, positive, and significant relationship between communication satisfaction and task performance [56,57,58,59], contextual performance [59], and overall work performance [44,52].

However, the relationship between communication satisfaction and work performance seems to be mediated by work engagement. This means communication satisfaction indirectly affects performance through the direct effect it has on work engagement [51,60,61,62], which, in turn, significantly predicts work performance [54].

### 1.4. Employee Work Engagement

As mentioned in the previous subsection, employee work engagement is one of the factors that has been repeatedly pointed out as a possible immediate antecedent of work performance [5,24]. This construct has been conceptualized in different ways, but the predominantly adopted definition is that proposed by Schaufeli et al. [63,64]. These authors consider work engagement as a special case of overall engagement [24,65,66]. They define it as a positive and satisfactory state of mind in relation to the daily activities that take place at work, characterized by vigor, dedication, and absorption [41,63,66,67,68]. It is a state that, while relatively stable over time, can change depending on what happens during the day [43,67].

According to the Work Engagement Model [67], the Job Demands–Resources Theory [8], and the HERO Model [7,9], employee work engagement leads to high performance. This relationship has been supported by empirical evidence obtained in different countries, which predominantly shows that employee work engagement significantly predicts organizational outcomes such as: (a) task/in-role performance [31,43,64,69,70,71,72,73]; (b) organizational citizenship behavior, also known as contextual performance or extra-role performance [43,69,70,71,72] (c) the helping behavior directed at co-workers [74]; (d) counterproductive work behavior [69]; (e) overall work performance [51,75,76,77]; and (f) effectiveness at the individual and organizational level [32,71], and at the team level [32,74].

A model that integrates empirical evidence regarding the possible direct and indirect relationships between leader–member exchange quality, communication satisfaction, employee work engagement, and self-rated work performance has been developed in this study. Such a model was tested (Figure 1) using mediation structural equation modelling.

In mediation models, a mediating variable (MV) is defined as the mechanism through which an independent variable (IV) affects a dependent variable (DV). Evaluating whether a variable acts by mediating the relationship between two other variables implies verifying that: (1) there is a statistically significant relationship between the IV and the DV; (2) variations in the IV significantly predict variations in the MV; (3) the MV significantly predicts the DV; (4) the IV–DV relationship either becomes significantly lower once the MV is included within the model—partial mediation—or the IV–DV relationship is no longer statistically significant once the MV is included within the model—full mediation [78].

Thus, evaluating the adequacy of the proposed mediation model (Figure 1) involved contrasting the following hypotheses:

 **Hypothesis 1 (H1).**
*The LMX quality–work performance relationship is mediated by communication satisfaction and employee work engagement. This means the direct relationship between both variables is either no longer statistically significant or decreases in magnitude when the mediation variables “communication satisfaction” and “employee work engagement” are considered. Thus, the confirmation of this hypothesis implies confirming that:*


(a)*The LMX quality evaluated by collaborators has a positive impact on work performance*.(b)*Employee work engagement has a positive impact on work performance*.(c)*Communication satisfaction of collaborators has a positive impact on work performance*.

 **Hypothesis 2 (H2).**
*The LMX quality–work engagement relationship is mediated by communication satisfaction, so that the direct relationship between both variables is either no longer statistically significant or decreases in magnitude when the mediation variable “communication satisfaction” is considered. Thus, the confirmation of this hypothesis implies confirming that:*


(d)*The LMX quality evaluated by collaborators has a positive impact on communication satisfaction*.(e)*The LMX quality evaluated by collaborators has a positive impact on employee work engagement*.(f)*Communication satisfaction of collaborators has a positive impact on employee work engagement*.

## 2. Materials and Methods

### 2.1. Participants

This study involved 488 workers (76.9% women) from public and private organizations, mostly from the education sector (85.7%), and primarily located in Ecuador (88.7%). Participants were between the ages of 17 and 75 years (M = 38.93; SD = 9.89). Most of them had college-level education (49.5% had a university degree, and 44.1% had completed postgraduate studies). The majority also worked full-time (90.3%) under a fixed-term or indefinite contract (77.3%) and had more than five years of work experience (81.8%) (Table 1).

This research has been reviewed and approved in January 2021 by the committee appointed for this purpose by the Research Center of Universidad Espíritu Santo, which has certified that this research conforms to the basic ethical norms of research. Following the ethical principles of psychologists and the code of conduct of the American Psychological Association [79], participation in this study was voluntary, and all subjects were provided with a digital informed consent after receiving information about: (a) the purpose of the research, expected duration, and procedures; (b) their right to decline to participate and to withdraw from the research once participation has begun without consequences; and (c) whom to contact for questions about the research. Given the characteristics of this research, participation did not imply any type of physical or psychological risk for the participants. Nor did it imply subjecting the participants to conditions that could cause them discomfort or that could have adverse effects on them. All data were treated confidentially, and were not shared with anyone other than the study authors. No specific incentive was used for participation in this study.

### 2.2. Instruments

#### 2.2.1. Work Performance

The Spanish version of the Individual Work Performance Questionnaire (IWPQ-version 1.0) [16] was used for this research. This instrument was selected since it was developed following a very rigorous process both in terms of the literature review that supported the selection of the items [80] and in terms of the different tests conducted by its original authors [14,15,18,81,82], as well as by other researchers [13,16,80,81,83,84], to evaluate its psychometric behavior.

The IWPQ version 1.0 is made up of 18 items that evaluate three dimensions of the construct: (a) task performance (5 items); (b) contextual performance (8 items); and (c) counterproductive work behaviors (5 items).

Task performance is defined as the behaviors that directly or indirectly contribute to the organization’s technical core [19,20,21,85,86]. This means the competence with which individuals perform their core tasks related to their jobs [13,14,15,17,21], which are normally included in their job descriptions [11,80,87] or in their contracts. Contextual performance is understood as the individual behaviors that contribute to maintaining the organizational, social, and psychological environment in which task performance takes place, and that go beyond what is formally prescribed as work objectives [11,13,14,15,17,21,85,87]. Counterproductive work behavior is defined as behaviors that harm the organization’s welfare [14,15,17,80,88,89].

In the IWPQ version 1.0, participants are asked to indicate how often they performed each of the behaviors indicated in the items in the last three months, using a Likert scale. The items of the task and contextual performance subscales are positive/directly worded. This means that the more frequently the indicated behaviors are performed, the better the work performance is. On the contrary, the items of the counterproductive work behavior subscale are negative/inversely worded, which means that the more frequently such behaviors are performed, the worse the work performance is.

The original five-interval scales (seldom, sometimes, frequently, often, and always (for the task and contextual performance items); and never, seldom, sometimes, frequently, and often (for the counterproductive behaviors items)) of the IWPQ used in this research were modified by using six-interval scales with the same response options for all items (1 = never/almost never–6 = almost always/always) and by labeling only the extremes of the categories. This was done because individuals perceive the original response options as unclear, especially when differentiating between “frequently” and “often” [15]. On the other hand, only the extremes of the scales have been labeled, since it has been observed that labelling all response options can lead to greater acquiescence bias [90]. Scales with an even number of intervals were used in order to reduce the probability that individuals responded systematically in the intermediate point, thus “compelling them” to select an option leaned more towards either extreme of the continuum. Furthermore, increasing the number of intervals of the scales increases the probability that data distribution fit to normal distribution and this may also contribute to improving the reliability of the instrument and its convergent validity [91].

#### 2.2.2. Leader–Member Exchange Quality

The Spanish version of the LMX–Multidimensional Scale (LMX–MDM) [36] was used for the purpose of this research. LMX–MDM is the scale most frequently used to measure leader–member exchange quality, and it was developed via a rigorous psychometric scale development procedures [92].

This instrument is made up of 12 statements regarding the interactions between employees and their immediate supervisors, from the employees’ perspective. It evaluates four dimensions: (a) affect, (b) loyalty, (c) contribution, and (d) professional respect.

Affect refers to the personal connections that dyad members have, which are based on interpersonal attraction rather than on work or professional aspects. Loyalty refers to public support expressions for the goals and personal characteristics of the other dyad member. Contribution deals with the actions carried out by each dyad member, aimed at achieving mutual objectives. Finally, professional respect means the extent to which dyad members consider their counterparts to have an excellent reputation in terms of their work field, either inside or outside the organization [25,36].

In LMX–MDM, each of those dimensions is measured by three items, and people are asked to indicate the extent to which they agree with each proposition, using a six-interval Likert scale (1 = strongly disagree–6 = strongly agree).

A reverse translation process was carried out to translate this instrument into Spanish. First, two independent bilingual English > Spanish translators translated the items into Spanish. Such translations were then compared by the researchers, who solved discrepancies based on the conceptual definitions of each of the instrument dimensions, as defined by its original authors. This first version was later translated into English by three other independent bilingual Spanish > English translators. These three versions were then compared with each other and with the original version in order to solve existing discrepancies and to guarantee conceptual and semantic equivalence between versions. In this way, the Spanish version was developed based on the consensus between the researchers and the translators who took part in this process.

#### 2.2.3. Communication Satisfaction

The Spanish version of the Communication Satisfaction Questionnaire (CSQ) [93] was adapted for the purpose of this research. The CSQ is considered the most comprehensive instrument for measuring different aspects of internal communication within organizations [93], and it is the most widely used for evaluating communication satisfaction in the organizational context [45,94,95,96].

The CSQ evaluates eight dimensions: (a) personal feedback, (b) supervisory communication, (c) informal communication or horizontal communication, (d) organizational integration, (e) organizational perspective, (f) communication climate, (g) media quality, and (h) communication with senior managers [97].

Personal feedback refers to the extent to which individuals perceive they are informed about their performance and the criteria they are being evaluated with [60,98]. It also refers to the extent to which supervisors understand the problems their employees deal with [97].

Supervisory communication refers to the satisfaction experienced by employees regarding the communication—downward and upward—they keep with their immediate supervisors [97]. This dimension focuses on whether supervisors show openness to new ideas, listen and pay attention, and on whether they guide and support their team members at work. Moreover, it includes the trust level between employees and supervisors [97].

Informal communication or horizontal communication refers to the extent to which this communication is fluid and the messages conveyed are accurate [60,97]. Organizational integration alludes to the satisfaction experienced by employees regarding the information they receive about their immediate work environment or unit [60,97]. Organizational perspective concerns communication with respect to the organization as a whole (e.g., news about changes, financial status of the organization, policies, and outcomes) [97]. Communication climate reflects the level of communication satisfaction at both the personal and organizational level [97], for instance, the extent to which communication motivates employees to achieve organizational objectives, and whether the attitudes of the members of the organization towards communication are healthy [60]. Media quality refers to the perception of employees regarding the effectiveness of the means of communication used in the organization [97]. Finally, communication with senior managers refers to the communication between the institutional level and the collaborators of the organization. This dimension has to do with the openness of managers towards the generation of new ideas and their concern for the welfare of the individuals that make up the organization [98].

This instrument was adapted based on the results obtained by Ancín-Adell [47]. This author included two questions in her study, in which participants could openly respond about communication satisfaction, and she also performed a textual content analysis to find out whether there were communication aspects related to satisfaction that were not addressed in the original CSQ. The results suggested the possible existence of an additional dimension: discourse tone. Such dimension is conceived as the degree to which communications between the members of an organization are characterized for being respectful, kind, and motivating [47].

Thus, the adapted version of the CSQ used in this study consisted of 49 items: five for each of the eight internal communication aspects considered in the original version of the instrument, and nine items regarding “discourse tone”. In each proposition, participants had to answer how satisfied they were with the amount and/or quality of the information referred to in each item, using a six-interval Likert scale (1 = strongly dissatisfied–6 = strongly satisfied).

#### 2.2.4. Employee Work Engagement

The official Spanish version of the Utrecht Work Engagement Scale (UWES-9) [99] validated by different authors in different countries was used [98,99,100,101,102]. The UWES is an instrument widely used worldwide to measure employee work engagement, and it is considered the standard measure of this construct [101,103,104,105,106,107,108,109].

The UWES-9 is made up of nine items; three for each dimension of employee work engagement: (a) vigor, (b) dedication, and (c) absorption [63]. Vigor refers to high levels of physical energy and mental endurance during work, the desire of investing effort into work, and persistence when facing difficulties. Dedication refers to the extent to which people feel strongly involved in their work, enthusiastic, and proud, and experience this as something meaningful. Absorption has to do with the extent to which employees are so focused and absorbed by work that time passes quickly for them and they have trouble disconnecting from it [24,41,63,65,66,67,68,104].

In the UWES-9, people must answer how often they have felt the way as expressed in each item, using a six-interval Likert scale (1 = never/almost never–6 = almost always/always).

### 2.3. Data Analysis

In order to evaluate the structural validity of each instrument used, a confirmatory factor analysis (CFA) was performed, whereas structural equation modeling (SEM) was conducted to test the proposed mediation model. The Lavaan package [110] in R Project for Statistical Computing (version 4.1.0) and the diagonally weighted least squares estimator (DWLS)—using a poly-choric correlation matrix—were used for this purpose. This estimator is recommended for analyzing samples with a moderate number of observations with ordinal data, such as that yielded by Likert-type items [111,112].

The χ2 tests, the root mean square error of approximation (RMSEA), the standardized root mean square residual (SRMR), the comparative fit index (CFI), and the Tucker–Lewis index (TLI) were used for model fit evaluation. The following criteria were used to evaluate model fit: RMSEA < 0.08, SRMR ≤ 0.08, CFI and TLI ≥ 0.95 [113], and χ2/*df* < 5 [114].

SPSS 25.0 was used for the descriptive analysis of the variables.

## 3. Results

### 3.1. Validation of Factorial Structure of the Instruments

#### 3.1.1. Individual Work Performance Questionnaire

Regarding the psychometric properties of the IWPQ, the results obtained in this study show adequate fit on the three-factor model (task performance, contextual performance, and counterproductive work behaviors) in all fit indices used (χ2/*df* = 2.889; CFI = 0.993; TLI = 0.992; RMSEA = 0.063, 90% CI = 0.055, 0.070; SRMR = 0.058), except for χ2, which was statistically significant (χ2 = 380.848, *df* = 132, *p* < 0.001). The inadequate goodness of fit, according to χ2, could be due to the fact that this test is very sensitive to detection of model misfit when working with large samples [114]. Figure 2 shows the factorial weights of the items in their corresponding factors.

In terms of reliability, adequate internal consistency indices were found for all dimensions (task performance: α = 0.834; contextual performance: α = 0.869; counterproductive work behavior: α = 0.820).

#### 3.1.2. LMX–Multidimensional Scale

Regarding the factorial structure of the Spanish translation of the LMX–Multidimensional Scale, carried out for the purpose of this study, the CFA evidenced adequate fit of the four-factor model (affect, loyalty, contribution, and professional respect) in all fit indices (χ2/*df* = 2.144; CFI = 0.999; TLI = 0.999; RMSEA = 0.049, 90% CI = 0.036, 0.062; SRMR = 0.031), except for χ2 (χ2 = 102.930, *df* = 48, *p* < 0.001). Figure 3 shows the factorial weights of the items in their corresponding factors.

Regarding reliability, high internal consistency was found for the affect (α = 0.944), loyalty (α = 0.897), and professional respect (α = 0.916) factors. However, the reliability of the contribution factor (α = 0.671) was of lesser magnitude.

#### 3.1.3. Communication Satisfaction Questionnaire

In terms of the factorial structure of the CSQ (adapted for the purpose of this study) the CFA evidenced adequate fit of the nine-factor model on three of the indices used (CFI = 0.995; TLI = 0.994; SRMR = 0.057). However, this was not the case for χ2 (χ2 = 6161.111, *df* = 1091, *p* < 0.001), χ2/*df* (χ2/*df* = 5.647), and RMSEA (RMSEA = 0.102, 90% CI = 0.099, 0.104). Figure 4 shows the factorial weights of the items in their corresponding factors.

The mismatch observed in RMSEA derives from the high correlation between factors (Table 2). Since it is expected that there will be some specification problems in such complex structures, maintaining the nine-factor theoretical structure was considered appropriate.

High internal consistency was found in all nine dimensions of the adapted version of the CQS: media quality: α = 0.932; communication climate: α = 0.911; supervisory communication: α = 0.924; informal communication or horizontal communication: α = 0.903; organizational integration: α = 0.858; organizational perspective: α = 0.888; personal feedback: α = 0.871; communication with senior managers: α = 0.958; discourse tone: α = 0.937.

#### 3.1.4. Utrecht Work Engagement Scale

Regarding the multidimensional structure of the UWES-9, the results obtained in the CFA evidenced an adequate fit of the three-factor model on the three indices used (CFI = 0.997; TLI = 0.996; SRMR = 0.042). However, this was not the case for χ2 (χ2 = 380.848, *df* = 132, *p* < 0.001), χ2/*df* (χ2/*df* = 6.115), and RMSEA (RMSEA = 0.103, 90% CI = 0.087, 0.119). As in the case of the CSQ, this mismatch observed in RMSEA derives from the high correlation between factors (r_vigor–dedication_ = 0.944, *p* < 0.001; r_vigor–absorption_ = 0.874, *p* < 0.001; r_dedication–absorption_ = 0.918, *p* < 0.001). Therefore, maintaining the theoretical structure was considered appropriate. Figure 5 shows the factorial weights of the items in their corresponding factors.

In terms of reliability, adequate internal consistency was found for all factors (vigor: α = 0.882; dedication: α = 0.872; absorption: α = 0.795).

### 3.2. Descriptive Analysis of the Variables

Overall, participants reported high levels of work performance, employee work engagement, communication satisfaction, and they considered that the relationship with their immediate supervisors was of high quality (Table 3). In this sense, all distributions presented negative skewness, thus indicating that the scores tended to cluster towards the high values of the scales (Table 3). The distributions of work performance, communication satisfaction, and leader–member exchange quality were mesokurtic, whereas that of employee work engagement was leptokurtic (Table 3). Finally, the datasets for work performance and employee work engagement were homogeneous, while those for communication satisfaction and leader–member exchange quality were heterogeneous (Table 3).

### 3.3. Testing of the Mediation Model

Adequate fit of the model tested (Figure 6) was found in all indices used (χ2/*df* = 1.044; CFI = 0.999; TLI = 0.999; SRMR = 0.056; RMSEA = 0.010, 90% CI = 0.003, 0.014), except for χ2 (χ2 = 3876.996, *df* = 3715, *p* = 0.031).

As for the prediction of work performance, the data support H1, according to which the relationship between leader–member exchange quality and work performance is mediated by communication satisfaction and employee work engagement. In this sense, all conditions indicated by Baron and Kenny [78] were met in this study so as to affirm that complete mediation occurred. Thus, a direct and significant relationship was found between leader–member exchange quality and work performance (a) β = 0.331, *p* < 0.001), which ceased to be statistically significant (β = 0.028, *p* = 0.218) when simultaneously considering employee work engagement—which significantly predicted work performance (b) β = 0.725, *p* < 0.001)—and communication satisfaction—which also significantly predicted work performance (c) β = 0.077, *p* < 0.001), although to a lesser extent.

Besides, and also as expected in H2: (d) leader–member exchange quality significantly predicted communication satisfaction (β = 0.684, *p* < 0.001). Finally, employee work engagement was significantly predicted by leader–member exchange quality (e) β = 0.260, *p* < 0.001) and by communication satisfaction (f) β = 0.431, *p* < 0.001), and the weight of the direct relationship between leader–member exchange quality and work engagement (β = 0.501, *p* < 0.001) was reduced when the mediating variable “communication satisfaction” was considered. Thus, the results herein show that, as expected (H2), the relationship between leader–member exchange quality and employee work engagement is partially mediated by communication satisfaction.

Table 4 presents a summary of the hypotheses tested and whether or not they were confirmed.

## 4. Discussion

The purpose of this study was to evaluate the fit of a mediation model in which the expected direct and indirect relationships between leader–member exchange quality, communication satisfaction, employee work engagement, and self-rated work performance are represented, based on what has been stated in the theoretical and empirical literature regarding the factors that affect work performance.

The results of this study show that the instruments used to measure the different variables are also valid and reliable in the case of samples gathered from Ecuadorian employees. Thus, in the case of the IWPQ, the findings of this research agree with those reported by other authors [80], since they have evidenced adequate fit of the three-factor model (task performance, contextual performance, and counterproductive work behavior). High internal consistency of all factors has also been confirmed, which agrees with what has been reported by different authors in other studies conducted in different countries using different language versions of the IWPQ [13,15,16,80,81,83,84].

Regarding the Spanish version of the LMX–MDM, the data obtained in this study evidenced adequate fit of the four-factor model (affect, loyalty, professional respect, and contribution), which agrees with what has been reported in the literature [25,33,92]. In terms of reliability, the internal consistency coefficients found in this study overall agree with those found in other studies [25,33,36,92].

Regarding the Spanish version of the CSQ (adapted for the purpose of this research), the nine-factor model (media quality, communication climate, supervisory communication, informal communication or horizontal communication, organizational integration, organizational perspective, personal feedback, communication with senior managers, and discourse tone) also showed adequate fit. Furthermore, each of the eight original dimensions of the CSQ had high internal consistency, which agrees with what has been found by other authors [47,94].

With respect to the UWES-9, the results herein agree with those found by other authors [100,101,106,107,115,116], thus proving adequate fit of the three-factor model (vigor, dedication, and absorption). Likewise, all factors had adequate internal consistency, which agrees with what has been reported in the literature [101,103,104,106,116,117,118,119,120,121].

Regarding the mediation model developed in this study, in line with the hypotheses herein, the results showed the existence of a significant relationship between exchange quality—or work and social interactions that are established between supervisors and supervisees—and some behaviors of employees that are relevant to the achievement of organizational objectives. Hence, as established in (a), employees consider their work performance to be higher as they perceive the relationship with their immediate supervisors is of higher quality. This relationship is consistent with what has been derived from the HERO Model [7,9], the Leader–Member Exchange Theory, and the principle of reciprocity of the Social Exchange Theory, as well as with what has been found in different research [28,34,37,39,42]. In this sense, employees with whom supervisors have high-quality relationships show better work performance, because in these types of relationships leaders provide their followers with more resources (e.g., autonomy, training and development opportunities, task variety, etc.), and there is a greater and more open exchange of valuable information, all within a climate of trust, sympathy, cooperation, consideration, and respect [29,30,31,32,33,34,36,37,38,39,40]. In other words, leaders treat subordinates with whom they have high-quality LMX more favorably than those with whom they have low-quality LMX [25,37]. Therefore, it is expected that the first show higher work performance levels [31,40].

The results of this study thus show that, as expected (H1), the relationship between LMX quality and work performance is completely mediated by employee work engagement and communication satisfaction. In this sense, coinciding with what has been proposed in the Engagement Model [37], the Job Demands–Resources Theory [8], the HERO Model [7,9], and what has been observed in different studies [31,32,38,43,51,64,69,70,71,72,73,74,75,76,77], employee work engagement significantly predicted work performance (b), employees consider their work performance to be higher as they feel more vigorous, involved and focused, and absorbed by work.

Employee work engagement leads to high performance probably because employees with high engagement levels experience positive emotions that encourage them to constantly work on improving their personal resources. They also experience better health conditions, which helps them focus and use their skills and resources on their jobs, create their own work, and convey their engagement with other people within their near environment [122].

As stated in (c), communication satisfaction also significantly predicted work performance, which is consistent with what has been proposed in the HERO Model [7,9] and by other authors [37,50,53], as well as with what has been reported in various empirical research [44,52,54,56,57,58,59]. Thus, employees claim they have better work performance as their satisfaction with respect to the communication environment of the organization increases.

When these two variables (work engagement and communication satisfaction) are considered, the direct relationship between leader–member exchange quality and work performance ceases to be statistically significant, which is evidence of the mediating role played by work engagement and communication satisfaction in the LMX quality–work performance relationship.

As for the prediction of employee work engagement, the results confirm (d), which states that communication satisfaction is significantly predicted by leader–member exchange quality. This is consistent with the Leader–Member Exchange Theory and with what has been found in other studies [29]. Likewise, as proposed in (e), and in line with the Job Demands–Resources Theory [8], the HERO Model [7,9], the Leader–Member Exchange Theory, and with what has been previously reported in the literature [30,31,32,38,43], employee work engagement was also significantly predicted by leader–member exchange quality. Finally, just as expected (f), employee work engagement was significantly predicted by communication satisfaction, which is consistent with the HERO Model [7,9] and agrees with what has been reported in other research [51,60,61,62].

Thus, this study shows that, just as expected (H2), the relationship between leader–member exchange quality and employee work engagement is partially mediated by communication satisfaction. That is, as employees consider the relationship with their immediate supervisors to be of higher quality, their communication satisfaction increases, and, as communication satisfaction increases, the engagement level of employees also increases.

Taken all together, the results herein indicate that leader–member exchange quality is relevant to the understanding of self-rated work performance, given the direct impact it has on communication satisfaction and because of the direct and indirect effect it has on employee work engagement, which, in turn, has a significant and positive effect on work performance.

Hence, these results represent additional empirical support to the approaches of the Job Demands–Resources Theory [8], the HERO Model [7,9], and the Leader–Member Exchange Theory, thus contributing to explaining and predicting work performance. Beyond that, this model allows understanding of how the direct and indirect relationships between relevant variables occur, taking from these theoretical perspectives, but whose relationships have been addressed separately. In this sense, this model demonstrates the relevance of the quality of the interactions that are established between immediate supervisors and supervisees, and of the satisfaction with organizational communication the latter experience, as resources and healthy work practices that have a positive impact, thus increasing employee work engagement. This, in turn, results in desirable organizational outcomes, such as having employees who exhibit high work performance levels.

From a practical point of view, the results herein suggest that, when willing to work towards developing healthier organizations, it is necessary to pay special attention to leaders or immediate supervisors, emphasizing what characterizes the interactions they establish with their supervisees. Hence, it is also necessary to implement recruiting and training practices aimed at increasing the number of employees with whom supervisors have high-quality relationships. Likewise, it is necessary to invest in adequate internal organizational communication strategies (downward, upward, and horizontal), so that the members of organizations feel satisfied with communication. Investing in this kind of resources and practices—despite not being the only options that lead to high individual, group and organizational performance—has a positive impact on employee work engagement and work performance.

Among the limitations of this study, it is relevant to point out that the sample was primarily composed of employees from the education sector, and that most of them had college-level education. Hence, it is necessary to be cautious when using the results presented herein and generalizing them for populations from other economic sectors and with lower educational backgrounds.

On the other hand, regarding the evaluation of the variables included in the model, it is not possible to rule out that at least part of the results that show that all distributions had negative skewness may be due to biases or response styles that represent sources of error variance, and that are recognized as a potential problem in social sciences, and particularly in behavioral research in the organizational context [123,124]. Among such biases, social desirability may have been present in this study. This refers to people’s tendency to adjust their responses to what they consider to be socially desirable. Another bias that may have occurred, particularly when measuring work performance, is acquiescence. This refers to the tendency to systematically respond in the affirmative or to agree with what is stated in the items regardless of their substantive content [125,126,127,128,129,130]. Acquiescence is a type of bias that typically occurs when using self-report measures [20] that employ Likert-like scales [125,130,131] with directly and inversely worded items, as is the case of the IWPQ.

## 5. Conclusions

Based on the approaches of the Job Demands–Resources Theory, the HERO Model, the Leader–Member Exchange Theory, and the empirical evidence found by different authors, the results herein lead to the following conclusions:The relationship between leader–member exchange quality and self-rated work performance is completely mediated by employee work engagement and communication satisfaction. This means that leader–member exchange is relevant to the understanding of work performance given its direct and positive impact on communication satisfaction, and because of its direct and positive effect on employee work engagement, which, in turn, has a significant and positive effect on work performance.The relationship between leader–member exchange quality and employee work engagement is partially mediated by communication satisfaction. This means that the magnitude of the direct relationship that exists between leader–member exchange and employee work engagement decreases when considering communication satisfaction.The latter implies that, as the members of an organization consider the relationship with their immediate supervisors to be of higher quality, the satisfaction they experience in terms of organizational communication increases, and as communication satisfaction increases, the extent to which employees feel more vigorous, involved and concentrated, and absorbed by work also increases, which, in turn, leads them to consider their work performance to be higher.

## Figures and Tables

**Figure 1 ijerph-19-08761-f001:**
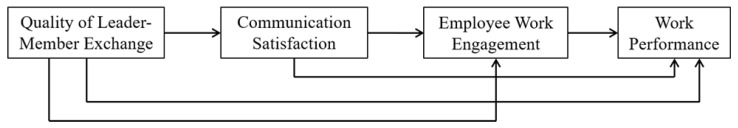
Proposed mediation model.

**Figure 2 ijerph-19-08761-f002:**
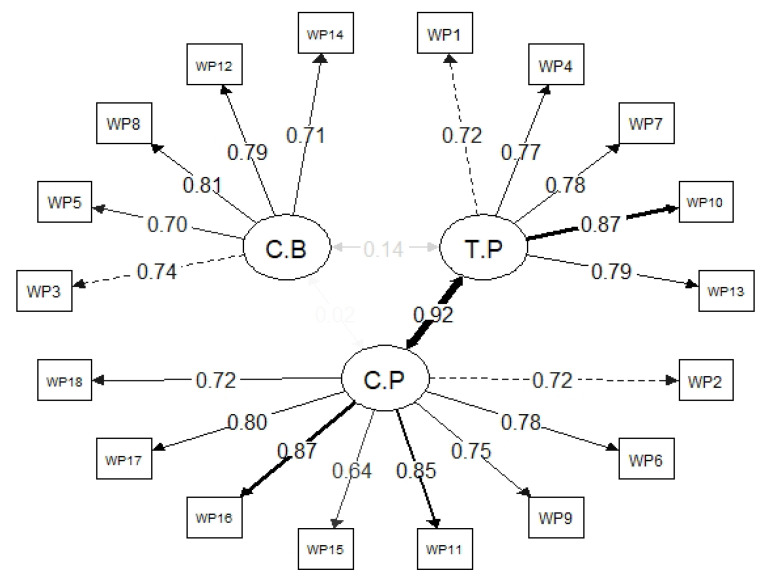
Factor loadings of the Individual Work Performance Questionnaire items. T.P = task performance. C.B = contextual performance. C.P = counterproductive work behaviors.

**Figure 3 ijerph-19-08761-f003:**
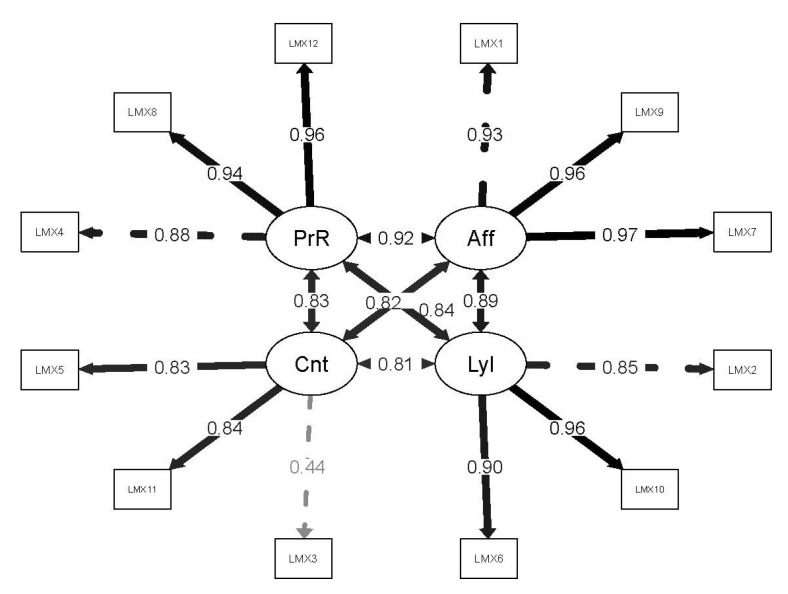
Factor loadings of the Spanish translation of the LMX–Multidimensional Scale items. Aff = affect. Lyl = loyalty. Cnt = contribution. PrR = professional respect.

**Figure 4 ijerph-19-08761-f004:**
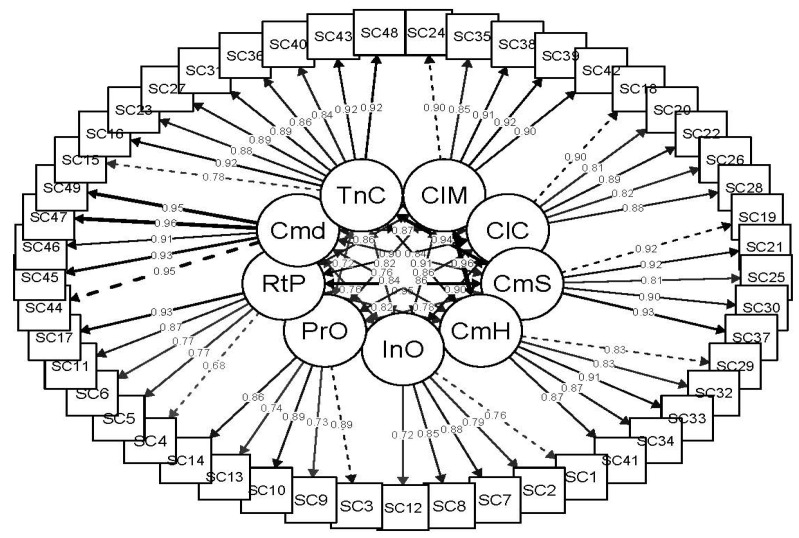
Factor loadings of the items of the adapted version of the Communication Satisfaction Questionnaire.

**Figure 5 ijerph-19-08761-f005:**
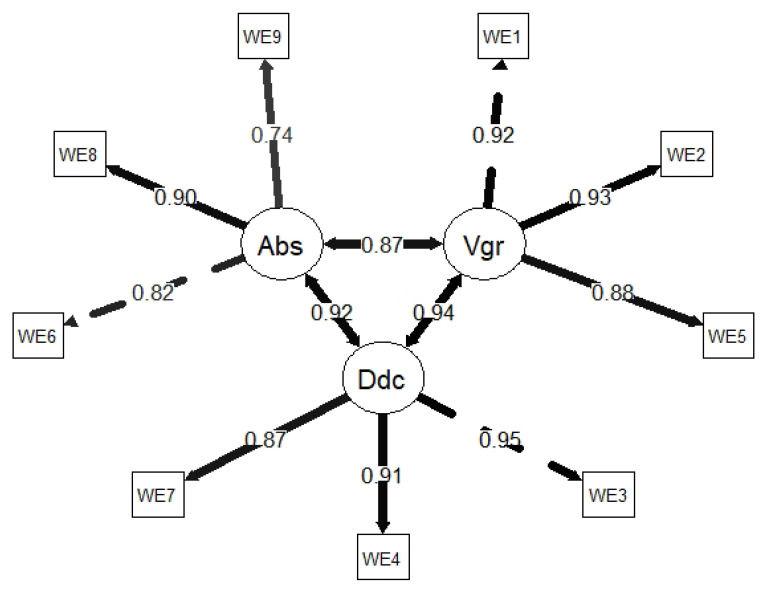
Factor loadings of Utrecht Work Engagement Scale items. Abs = absorption; Vgr = vigor; Ddc = dedication.

**Figure 6 ijerph-19-08761-f006:**
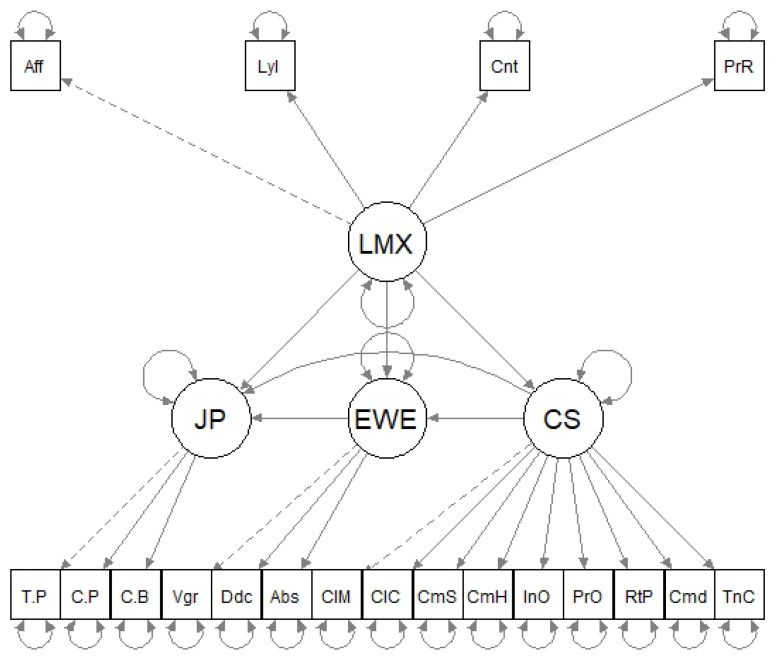
Mediation model tested. LMX = leader–member exchange. JP = job/work performance. EWE = employee work engagement. CS = communication satisfaction.

**Table 1 ijerph-19-08761-t001:** Sample characteristics.

		Percentage
Gender	Male	23.1
	Female	76.9
Education level	Elementary or middle school	0.4
	Community college degree	0.4
	High School	4.6
	College degree	49.5
	Postgraduate degree	44.1
	Other	1.0
Years of work experience	Less than 1 year	1.5
	Between 1 and 5	16.7
	More than 5 years	81.8
Time in the organization	Less than 1 year	15.6
	Between 1 and 5	39.2
	More than 5 years	45.2
Time in current position	Less than 1 year	19.1
	Between 1 and 5	44.4
	More than 5 years	36.5
Type of contract	Fixed-term or indefinite contract	77.3
	Civil servant contract	5.9
	Temporary contract	8.6
	Freelance	2.1
	Employed without a contract	0.2
	Other	5.9
Type of employee	Part-time	9.7
	Full-time	90.3
Type of organization	Public	30.1
	Private	69.9

**Table 2 ijerph-19-08761-t002:** Correlations between factors of the adapted version of the communication satisfaction questionnaire.

	1	2	3	4	5	6	7	8	9
1 CIM	1.000	0.984	0.920	0.960	0.861	0.824	0.860	0.910	0.899
2 CIC		1.000	0.926	0.940	0.898	0.857	0.919	0.869	0.875
3 CmS			1.000	0.860	0.824	0.777	0.951	0.836	0.944
4 CmH				1.000	0.829	0.800	0.816	0.841	0.911
5 InO					1.000	0.967	0.999	0.761	0.764
6 PrO						1.000	0.923	0.787	0.718
7 RtP							1.000	0.783	0.854
8 Cmd								1.000	0.893
9 TnC									1.000

CIM = media quality; CIC = communication climate; CmS = supervisory communication; CmH = horizontal communication; InO = organizational integration; PrO = organizational perspective; RtP = personal feedback; Cmd = communication with senior managers; TnC = discourse tone. All correlations are significant at *p* < 0.001.

**Table 3 ijerph-19-08761-t003:** Descriptive statistics of the variables.

Variables	M	Md	SD	CV	Skewness	Kurtosis
Job performance	4.942	5.000	0.599	12.12%	-0.424	−0.050
Employee work engagement	5.182	5.333	0.791	15.26%	−1.007	0.569
Communication satisfaction	4.616	4.700	0.937	20.12%	−0.771	0.302
Leader–member exchange quality	4.792	5.000	1.050	21.91%	−0.910	0.118

**Table 4 ijerph-19-08761-t004:** Hypotheses tested and results obtained.

Hypotheses	Results
H1: LMX → JP mediated by CS and EWE	Accepted
(a) LMX → JP	Accepted
(b) EWE → JP	Accepted
(c) CS → JP	Accepted
H2: LMX → EWE mediated by CS	Accepted
(d) LMX → CS	Accepted
(e) LMX → EWE	Accepted
(f) CS → EWE	Accepted

## Data Availability

The data presented in this study are available on request from the corresponding author.

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
