# Peer review of "Incidence of Leader–Member Exchange Quality, Communication Satisfaction, and Employee Work Engagement on Self-Evaluated Work Performance"

_ijerph, 2022, doi:10.3390/ijerph19148761_

Round 1

Reviewer 1 Report

Dear author(s),

First of all, thank you for the opportunity to review this manuscript.

I congratulate the authors for the detailed, thorough study. Please, see below some comments that can contribute to improve your work. Although most are minor issues, there is one important point that should be taken into account in the improved version at the end of this review.

General comments

·         The abstract is well written and contains the main elements of the paper. The chosen keywords complement the title and abstract, contributing to the study being found by interested researchers.

·         The introduction is well-structured and provides the necessary elements for the reader to understand the research gap and the aim of the authors. The authors show the importance of the theme in an adequate way, citing appropriate references.

·         The method is well explained and is suitable for both the proposed objectives and the existing research gap.

·         The results are consistent with the literature and the methodology.

·         The conclusions are supported by the results.

Issues to be considered in the revision of the manuscript

·         Please consider providing references for the following: “Organizations are social systems created and shaped by people with shared goals, norms, and values.” – There is many different ways of defining an organization, including some built upon conflicting conceptual basis, so it is important to indicate references for the reader to understand which concepts/theories the authors are building on.

·         Please consider providing references for the following: “These systems are structured in subsystems that must be coordinated in order to achieve organizational goals. Therefore, the performance of any organization depends fundamentally on the behavior and interrelationships that occur between its members, and between them and stakeholders” – There is many different ways of defining an organization, including some built upon conflicting conceptual basis, so it is important to indicate references for the reader to understand which concepts/theories the authors are building on.

·         I suggest that, although you have stated what “HERO” means in the abstract, it could also be done in the first use in the text (line 45)

·         Please consider the following sentence: “with a hierarchical structure with a general factor on its "tip", followed by several dimensions at the lower levels” – Is “tip” the right word?

·         Please consider referencing the Spanish version of the UWES-9 adequately, not in the text (i.e., lines 391-392 - https://www.wilmarschau-391 feli.nl/publications/Schaufeli/Tests/UWES_ES_S_9.pdf.). This should appear in the references section at the end of the manuscript.

·         If the manuscript progresses through the review process, please provide high-definition images.

·         Please consider the following statement (line 535): “The results of this study prove that the instruments used to measure the different variables are also valid and reliable in the case of samples” – Please consider replacing the word "prove"; prefer to use verbs that do not close the doors to contradictory, which is essential in any scientific study

·         The same should be considered in all the times that the word “prove” is used, such as in line 577: “The results of this study thus prove that, as expected (H1), the relationship between”; line 612: “Thus, this study proves that, just as expected (H2), the relationship between”; line 618: “Taken all together, the results herein prove that leader–member exchange quality”; line 634: “From a practical point of view, the results herein prove that when willing”. Please read the entire text carefully to correct this important issue whenever it appears, looking beyond the ones I have indicated.

Author Response

Dear revisor 1:

Thank you very much for dedicating part of your time to reviewing our manuscript and for your valuable suggestions. We have incorporated all your recommendations. Changes are highlighted within the manuscript. Please see below for a point-by-point response to the comments. All line numbers refer to the revised manuscript file.

  • Please consider providing references for the following: “Organizations are social systems created and shaped by people with shared goals, norms, and values.” – There is many different ways of defining an organization, including some built upon conflicting conceptual basis, so it is important to indicate references for the reader to understand which concepts/theories the authors are building on.
  • Please consider providing references for the following: “These systems are structured in subsystems that must be coordinated in order to achieve organizational goals. Therefore, the performance of any organization depends fundamentally on the behavior and interrelationships that occur between its members, and between them and stakeholders” – There is many different ways of defining an organization, including some built upon conflicting conceptual basis, so it is important to indicate references for the reader to understand which concepts/theories the authors are building on.

In the new version of our manuscript we have specified the theories that we have used as the basis for the conception of organization assumed in the research, as well as the respective references.

Although there is no univocal answer to the question, “what is an organization?,” following the basic notions of the Systems Theory, the Institutional Theory, and the Human Relations Theory [Griseri, 2013; Rivas-Tovar, 2009], organizations can be conceived as social systems created and shaped by people who, in some way, intend to work towards achieving similar or common objectives [Griseri, 2013; Rivas-Tovar, 2009; Robbins & Judge, 2013]. These systems are structured in subsystems that must be coordinated in order to achieve organizational goals [Griseri, 2013; Rivas-Tovar, 2009; Robbins & Judge, 2013]. Therefore, following the foundations of the Human Relations Theory [Rivas-Tovar, 2009], the performance of any organization depends fundamentally on the behavior and the interrelationships that occur between its members, and between them and stakeholders.

  • I suggest that, although you have stated what “HERO” means in the abstract, it could also be done in the first use in the text (line 45)

Following your suggestion, we have included the meaning of HERO the first time we refer to this model within the text (Line 49).

  • Please consider the following sentence: “with a hierarchical structure with a general factor on its "tip", followed by several dimensions at the lower levels” – Is “tip” the right word?

The word "tip" was replaced by “…at the top of the structure…” (Line 73)

  • Please consider referencing the Spanish version of the UWES-9 adequately, not in the text (i.e., lines 391-392 - https://www.wilmarschau-391 feli.nl/publications/Schaufeli/Tests/UWES_ES_S_9.pdf.). This should appear in the references section at the end of the manuscript.

The reference to the Spanish version of the UWES-9 was included in the references and removed from the text (see line 393).

  • If the manuscript progresses through the review process, please provide high-definition images.

Figures with a higher degree of resolution have been included in the new version of the manuscript.

  • Please consider the following statement (line 535/536): “The results of this study prove that the instruments used to measure the different variables are also valid and reliable in the case of samples” – Please consider replacing the word "prove"; prefer to use verbs that do not close the doors to contradictory, which is essential in any scientific study. The same should be considered in all the times that the word “prove” is used, such as in line 577/578: “The results of this study thus prove that, as expected (H1), the relationship between”; line 612/613: “Thus, this study proves that, just as expected (H2), the relationship between”; line 618/619: “Taken all together, the results herein prove that leader–member exchange quality”; line 634/635: “From a practical point of view, the results herein prove that when willing”. Please read the entire text carefully to correct this important issue whenever it appears, looking beyond the ones I have indicated.

The entire manuscript was revised in order to replace the verb "prove" with others that do not close the doors to contradictory

Reviewer 2 Report

The paper represents valuable and interesting results for the topic. Additionally, I would suggest, even if there was mentioned that the level of education of the respondents can be seen as limitation of the study, the authors could use both gender and education level as control variables, thus enhancing the internal validity of the results.

Author Response

Dear reviewer 2: Thank you very much for dedicating part of your valuable time to reviewing our manuscript and for your observations. In this sense, regarding the use of gender and educational level as control variables, we do not consider it appropriate to include these variables in the model because in this research there was great heterogeneity in the sample sizes of the possible sub-samples (for example, in the case of gender, 77% of the participants were female), so that the sample size in some sub-samples is below what is recommended for CFA and SEM models. For this reason, the results that could have been obtained by including these two variables as control variables would not be reliable, regardless of whether they suggested a significant effect or a non-significant effect. Larger sample sizes are required for some of the sub-samples so that the statistical power is high enough to correctly identify possible effects.

Reviewer 3 Report

The article is well written, therefore, what it wants to convey is well understood and in my opinion the work has enough information in the analysis of the results and discussion of the work; I only find some details regarding the observations of the article that are listed below.

Figure 2 and Figure 3 both are recommended to improve them, they look opaque and therefore they are not well appreciated.   Line 425 and 436 the values ​​that are written should have a zero before the period. Example 0.323   Table 2 the values ​​that are recorded should have a zero before the point.   Line 448, 449 and 453 have hyphens between the words, is that correct?

Author Response

Dear reviewer 3: Thank you very much for dedicating part of your valuable time to reviewing our manuscript and for your observations. We have incorporated all your recommendations. Changes are highlighted within the manuscript. Please see below for a point-by-point response to the comments. All line numbers refer to the revised manuscript file.

  • Figure 2 and Figure 3 both are recommended to improve them, they look opaque and therefore they are not well appreciated.

Figures with a higher degree of resolution have been included in the new version of the manuscript.

  • Line 425 and 436 the values ​​that are written should have a zero before the period. Example 0.323 Table 2 the values ​​that are recorded should have a zero before the point.  

Following the standards of the Publication Manual of the American Psychological Association (2020), zero had not been placed before the point in those cases in which the value of the statistic could not be greater than 1. However, following its recommendation, we have placed a “zero” before the point in all number.

  • Line 448, 449 and 453 have hyphens between the words, is that correct?

Yes, in those cases it is correct to use a hyphen between two words to express compound adjectives.